# Sex Hormone-Dependent Physiology and Diseases of Liver

**DOI:** 10.3390/ijerph17082620

**Published:** 2020-04-11

**Authors:** Paulina Kur, Agnieszka Kolasa-Wołosiuk, Kamila Misiakiewicz-Has, Barbara Wiszniewska

**Affiliations:** Department of Histology and Embryology, Faculty of Medicine and Dentistry, Pomeranian Medical University, Powst. Wlkp. 72, 70-111 Szczecin, Poland; paulina.kur@pum.edu.pl (P.K.); kamila.misiakiewicz@pum.edu.pl (K.M.-H.); barbara.wiszniewska@pum.edu.pl (B.W.)

**Keywords:** gender-dependent liver failure, hepatic glucose metabolism, insulin resistance, type 2 diabetes, metabolic syndrome, hepatic lipid metabolism, non-alcoholic fatty liver disease, cirrhosis, hepatocellular carcinoma, transgenic animal models, clinical cases

## Abstract

Sexual dimorphism is associated not only with somatic and behavioral differences between men and women, but also with physiological differences reflected in organ metabolism. Genes regulated by sex hormones differ in expression in various tissues, which is especially important in the case of liver metabolism, with the liver being a target organ for sex hormones as its cells express estrogen receptors (ERs: ERα, also known as ESR1 or NR3A; ERβ; GPER (G protein-coupled ER, also known as GPR 30)) and the androgen receptor (AR) in both men and women. Differences in sex hormone levels and sex hormone-specific gene expression are mentioned as some of the main variations in causes of the incidence of hepatic diseases; for example, hepatocellular carcinoma (HCC) is more common in men, while women have an increased risk of autoimmune liver disease and show more acute liver failure symptoms in alcoholic liver disease. In non-alcoholic fatty liver disease (NAFLD), the distinction is less pronounced, but increased incidences are suggested among men and postmenopausal women, probably due to an increased tendency towards visceral fat accumulation.

## 1. Sex Hormone-Dependent Glucose Metabolism in a Healthy Liver, in Insulin Resistance (IR) and in Diabetes (T2D)

### 1.1. Cellular Transporters Involved in Glucose Transport: Their Expression in IR, T2D and MetS (metabolic syndrome)

The liver is one of the organs responsible for glucose metabolism due to its production of glucose (glucogenesis), which is stored as glucogen (glycogenogenesis) (Figure 1), and is degraded as needed via the glycolytic pathway (glycogenolysis) or converted to fatty acids by the lipogenic pathway (lipogenesis). Glucose transport is directed by the Na^+^-coupled glucose transporters (SGLT) or glucose transporters (GLUT). Among the former, SGLT1 and SGLT2 serve as transporters, and SGLT3 is a glucose sensor. In humans, the GLUT family includes 14 isoforms which have diverse affinities and different expression profiles, thus enabling tissue adaptation of glucose uptake via *GLUT* gene expression [1]. Glucose absorption and release depends on the current needs of the body, and take place mainly through the activation of GLUT2, one of GLUT’s isoforms. Bi-directional transport via GLUT2 is responsible for the glucose balance in the cell, and GLUT2 up-regulation plays a more important role in the export of glucose than in its import to the liver [1]. Nevertheless, to ensure proper expression of glucose-dependent genes in the liver, it is necessary to maintain a proper balance between intracellular and extracellular GLUT2-dependent glucose concentrations [2]. In the liver, the GLUT2 level is increased by glucose, insulin and fatty acid synthase (FASN) stimulation, and studies on GLUT2 knockout mice confirm that glucose uptake by hepatocytes is a major source of glucose for lipogenesis [2].

Epidemiological studies show sex differences in type 2 diabetes (T2D) and indicate a higher prevalence in men than in women [3]. Men are also more likely to suffer from obesity and effects of sedentary lifestyle than women, probably due to differences in insulin sensitivity and regional fat storage [4], which may be due to disrupted sex hormone homeostasis. In a study on castrated rats, an increased level of glucose in blood and a higher level of GLUT2 mRNA and protein expression was found as a result of endogenous androgens deficiency [5]. Supplementation with testosterone (T) or testosterone with estradiol (E2) normalized the level of GLUT2 mRNA and protein expression in the liver of the rats, whereas supplementation with E2 alone had no effect [5]. In vitro data indicated that the addition of testosterone and 17β estradiol to the medium of non-malignant Chang liver cells significantly increased the insulin receptor mRNA expression and glucose oxidation and that these processes were not the effect of insulin action. So, compared to the individual effects of T and E2, their combination significantly increased the glucose oxidation which is similar to the effect of insulin [6]. In the aforementioned study by Muthusamy [5], testosterone deficient rats were characterised by increased hepatic glucose synthesis (hyperglycaemia) and symptoms similar to T2D or metabolic syndrome (MetS) [5]. Because the normal testosterone level improved the levels of GLUT2 mRNA and protein expression, it can be supposed that T directly influences the *GLUT2* gene transcription and translation [5]. Another report indicated that exogenous T encouraged synthesis of glycogen in both castrated and non-castrated rats [7]. At the same time, in humans, a high level of testosterone was related to a low risk of diabetes in men and a high risk in women [7]. An excess of androgens in women with polycystic ovary syndrome (PCOS) disrupts hepatic glucose metabolism as a result of a reduced glucose concentration in blood due to insulin action and glycogen synthesis; furthermore, PCOS predisposes women to insulin resistance (IR) [8,9]. However, estrogens have been shown to have little effect on GLUT2 and the insulin receptor in the livers of male rats, although this caused an increase in the insulin receptor levels in the human liver cell line (HepG2) [10] and non-malignant Chang liver cells [6]. Furthermore, testosterone supplementation resulted in the non-malignant Chang liver cells up-regulating the mRNA level for the insulin receptor and increasing insulin sensitivity [6].

### 1.2. Relationship between Androgens/Androgen Receptor/5α-Reductase and Hepatic Glucose Homeostasis

A lack of an androgen receptor (AR) in males promotes IR which could promote T2D development. For example, an in vivo study performed on AR knockout (*AR*^−/*y*^) male mice showed a gradual decrease in sensitivity to insulin and attenuated glucose tolerance which increased with age. Aging *AR*^−/*y*^ mice also exhibited accelerated hyperinsulinemia [4]. Another experiment on sex-dependent insulin resistance was carried out on hepatic AR knockout mice, where overweight male H-AR^−/y^, but not female H-AR^−/−^ mice, fed a high-fat diet (HFD) were characterized by reduced sensitivity to insulin as a result of increased expression of protein-tyrosine phosphatase 1B (PTP1B, negative regulator of the insulin signaling pathway). So, the hepatic androgen receptor, as a positive factor, could also play an important role in avoiding IR development [11]. The authors of the publication suggest that “strategies aimed at increasing AR activity specifically in the liver through tissue-selective AR modulators could therefore improve both hepatic insulin and leptin sensitivity and improve both lipid and glucose homeostasis”. In another study 5α-reductase-knockout (5αR1^−/−^, but not 5αR2^−/−^) mice with an ALIOS diet (American lifestyle-induced obesity syndrome) had decreased hepatic mRNA expression of genes involved in insulin signaling [12]. However, overweight male Zucker rats (castrated and non-castrated) showed hyperinsulinemia induced by finasteride (which is an inhibitor of 5α-R2 and not 5α-R1) [13]. Male 5αR1-knockout mice on HFD showed a higher average weight gain and hyperinsulinemia then wild type (WT) animals [13]. This suggests that lack of activity of 5α-reductase, the enzyme that converts T to DHT, induces IR. Interestingly, in another study [14], expression of the ERα transcript in the liver was decreased by DHT treatment of orchidectomized (ORX) male mice, although there was no significant impact on ERβ or AR transcripts.

### 1.3. Relationship between Estrogens/Estrogen Receptors/Aromatase and Hepatic Glucose Homeostasis

The relationship between estrogen concentration and metabolic homeostasis has also been found in a study carried out on aromatase-deficient patients and aromatase-knockout animals (ArKO mice); these patients, such as a male patient with inactive ERα [15], displayed diminished glucose metabolism, insulin resistance and hyperinsulinemia [16]. Exogenous estrogen supplementation of the male patient with attenuated ERα did not recover homeostasis of glucose and did not restore insulin to normal level, whereas the aromatase-deficient patients revised their metabolic anomalies [15]. Estrogen supplementation increases synthesis and release of insulin [17,18], and may also change liver GLUT expression [1]. Therefore, these sex hormones are important in hepatic insulin clearance. Postmenopausal women, after an oral hormone replacement therapy (HRT) at low doses, did not show any changes to IR but exhibited a slight increase in hepatic insulin clearance [19]. Estrogens also reduce gluconeogenesis and increase liver glycogen synthesis and storage, and lower blood glucose levels [20]. As well as this, observations on ovariectomized (OVX) rodents support the view that estrogens decrease glucose levels [21,22]. This correlates with increased glucagon signalling due to an increased expression of the glucagon receptor (GLR), which stimulates glucose production by activating gluconeogenic enzymes in OVX rats [23]. As a glucose imbalance after an ovariectomy is not reversed by exogenous E2 [23], it can be assumed that this disturbed homeostasis was due to a lack of progesterone. While the classic, nuclear progesterone receptor (PR) has not been detected in the liver [24], progesterone—in addition to binding to membrane PR [25]—can also bind to the nuclear AR [26] and thus induce metabolic effects in human hepatocytes, similar to hepatic glycogenolysis and gluconeogenesis regulated by the epinephrine/β2-adrenergic receptor pathway as a result of estrogen action [27].

T2D and MetS development are known to be related to the polymorphism in the gene encoding ERα; for example, a study on men with identified ERα-deficiency showed an imbalance in glucose metabolism [28]. ERα knockout (ERαKO) mice also display attenuated glucose tolerance with IR in their livers, while ERβ knockout (ERβKO) mice have normal glucose tolerance, which suggests that ERα and not ERβ plays an important role in regulating glucose homeostasis in the liver [20,29]. The normalization of glucose homeostasis, insulin concentration and the reverse obesity in mice with ERα deficiency and removed ovaries, suggests that ERβ may be a base of diabetogenic and adipogenic phenotype. In contrast, ERβKO mice had better insulin sensitivity and glucose tolerance without increased body fat storage. That is why ERα is indicated as an important factor in metabolic regulation [29]. Similarly, a declining glucose tolerance was also observed in GPER1 knockout mice (G protein-coupled estrogen receptor 1), although GLUT2 expression and glucokinase activity were not altered [30]. The mouse GLUT2 promoter includes both a PPAR-γ (peroxisome proliferator activator receptor-gamma) response element as well as a steroid hormone response element [31,32], and that is why ERs with PPAR-γ could together regulate the gene encoded glucose transporter type [33].

Ovariectomies of female Holstman rats did not affect liver glycogen levels or phosphoenolpyruvate carboxykinase (PEPCK, the main enzyme in the gluconeogenesis pathway) gene expression, but after E2 replacement the expression was altered (glycogen–increased, PEPCK–decreased) [34]. However, according to another study [35], following the ovariectomy of obese female rats the expression of lipogenic (sterol-regulatory element-binding protein 1c, SREBP-1c; FASN) and adipogenic (PPAR-γ) genes in the liver increased significantly, and administration of exogenous E2 or an ERα agonist (16a-LE2) reduced hepatic expression of SREBP-1c, FASN and PPAR-γ, while an ERβ agonist (8β-VE2) comparably increased PPARγ expression to the same level of mRNA as in non-treated ovariectomized animals [35]. Both agonist of ERs not only decreased lipogenesis but also lowered triglyceride (TG) accumulation in the liver. Therefore, the systemic insulin sensitivity was improved by the activation of ERα and also ERβ, most likely as a result of the anabolic activity of ERβ [35]. (Summary of the above data in Table 1.)

## 2. Sex Hormone-Dependent Lipid Metabolism in the Normal Liver, in Nonalcoholic Fatty Liver Disease (NAFLD), Obesity and in Metabolic Syndrome (MetS)

There is an increasing amount of evidence from animal cell culture and clinical studies that testosterone/androgens control the expression of important regulatory proteins involved not only in glycolysis and glycogen synthesis but also in lipid and cholesterol metabolism [7]. For example, dihydrotestosterone treatment of orchidectomized male mice caused obesity, decreased energy utilization and fat oxidation, ane increased HDL-C and TG levels correlating with lowered bile acid synthesis as a result of down-expression of 7α-hydroxylase gene [14]. In addition, in AR knockout male mice (*AR*^−/*y*^) the exogenous DHT did not reverse the metabolic anomalies and IR [11].

### 2.1. Relationship between Estrogens/Estrogen Receptors/Aromatase and Hepatic Lipid Homeostasis

In the case of hepatic lipid metabolism, it should be emphasized that estrogens are also crucial. In both sexes, estrogen signalling via ER is important in the regulation of lipogenesis, as evidenced by experimental animal models and clinical studies. Women with breast cancer were treated with tamoxifen (selective modulator of ER, with anti-estrogenic effect) although hepatic steatosis caused by an impairment of fatty acids (FA) β-oxidation in estrogen deficient livers is a frequent complication associated with this therapy [1]. Genome-wide analysis demonstrates that transcriptional activity of ERα oscillates depending on the phase of the mouse estrous cycle and this oscillation is required for pulsating transcription of FA and cholesterol genes. This ER-dependent physiological programming changes during gestation and after termination of fallopian tube as a result of progressing age or surgically induced menopause, signifying that ER signalling is crucial for appropriate liver physiology in relation to the energetic necessities of reproductive age. Therefore, any changes in the amplitude and frequency of the cycle are related with the accumulation of fat in the liver [36]. Accordingly, the oscillation of ER expression has great importance for limiting fat deposition in the livers of women of reproductive age, and appropriate HRT in post-menopausal women or after surgical menopause, and has an important role for hepatic metabolism [36]. This protective function of estrogens is mainly attributed to ERα signalling [37] because, as was shown in microarray analysis, male and female ERαKO mice exhibit a fatty liver due to the up-expression of lipogenic genes and down-expression of genes involved in lipid intake [34]. Mice with liver ERα-knockout (LKO) [38] and Gpr30-deficient mice (GPR30; orphan G protein-coupled receptor 30) [39] fed a HFD had increased liver triglycerides and diacylglycerides, and female rather than male mice had significantly lower HDL-C level along with an increase in fat liver accumulation with insulin resistance. Thus, both ERα and GPER (G protein-coupled ER, also known as GPR 30) must be present in the liver cells to maintain lipid homeostasis.

The estrogenic pathway of regulation of the liver function also heavily depends on the activity of aromatase, an enzyme that converts androgens to estrogens. In aromatase knockout (ArKO) male mice, but not in ArKO female mice, the developed liver steatosis could be normalized by the administration of exogenous estrogens [40], and impaired hepatic FA β-oxidation was caused by disturbing the activity of peroxisomal very long-chain acyl-CoA synthetase (VLACS), fatty acyl-CoA oxidase (AOX) and medium-chain acyl-CoA dehydrogenase [41,42]. These impairments were inverted by exogenous E2 [41] or treatment with pitavastatin that is able to re-establish FA β-oxidation via the PPAR-α and abolish hepatic steatosis [42]. As such, estrogen therapy is used to aid recovery from metabolic anomalies in aromatase-deficient patients [15]. In castrated male rats, E2 supplementation also decreased FA production and lipid collection, and prevented nonalcoholic fatty liver disease [43]. Similar to the increase in hepatic triglycerides and diacylglyceride in liver ERα-knockout HFD fed mice, the alteration of insulin-stimulated ACC (acetyl-CoA carboxylase) phosphorylation and DGAT1/2 (diacylglycerol O-acyltransferase 1/2) protein levels were also observed due to decreased insulin sensitivity [38]. Males on a HFD showed that estrogen (via ERα) helps avoid not only hepatic but also whole body IR. Therefore, intensifying hepatic estrogen-ERα pathways could reduce the effect of obesity, diabetes and cardiovascular risk [38]. Another study on a mouse knockout model (ArKO) [44] showed obesity and liver steatosis due to an impaired FA β-oxidation and an increased FASN level in the liver of both female and male mice. This is in line with the findings of Foryst-Ludwig et al. [45], according to whom ERα mainly mediates beneficial metabolic effects of estrogens such as anti-lipogenesis, improvement of insulin sensitivity and glucose tolerance, and reduction of body fat mass. In contrast, ERβ activation seems to be detrimental for the maintenance of regular glucose and lipid homeostasis. Hypoestrogenemia caused by ovarian senescence significantly increases the risk of steatohepatitis and liver fibrosis severity both in humans with NAFLD and in zebrafish with experimental steatosis [46]. Therefore, estrogen deficiency promotes the development of NASH (nonalcoholic steatohepatitis) and estrogen treatment improved NASH progression in bilateral ovariectomized mice fed a high-fat and high-cholesterol (HFHC) diet [47].

It was also shown that not only classic, nuclear ERα and ERβ act in liver metabolism. Some very interesting results are presented in a study carried out on a “membrane-only ERα” mouse model (MOER, ligand-binding domain of receptor is present within the plasma membrane) injected with propyl-pyrazole-triol or trisphenol (PPT), a selective agonist of ERα [48]. This experiment showed that the expression of many lipid synthesis-related genes was decreased in “MOER” mice but was not suppressed in ERKO mice, indicating that only membrane-localized ERα was necessary for the suppression of these genes (cholesterol, TG and FA content was decreased only in livers from MOER mice exposed to PPT, but not in the livers from the ERKO mice). Therefore, the inhibition of gene expression mediated by membrane-localized ERα caused the aforementioned metabolic phenotype that did not require nuclear ERα. Consequently, the membrane-localized ERα is responsible for protection against hyperlipidemia by reducing the expression of some genes involved in lipid synthesis in the liver [48]. Although ERα in the liver is considered anti-lipogenic, data from literature on the role of ERβ in the liver is not consistent. Mice with a lack of ERβ are heavier but their livers are lighter as a result of reduced hepatic TG storage accompanied by whole body higher insulin sensitivity [49], indicating that ERβ in the liver can perform lipogenic and diabetogenic functions, because—as was documented by Foryst-Ludwig et al. [49]—this receptor deactivates the adipocytic gene expression induced by PPAR-γ and finally leads to a reduction in adipogenesis. This is confirmed by the discoveries of some mutations in the ERβ gene of obese female adolescents or women with bulimia [50,51].

### 2.2. Correlation of Non-Alcoholic Fatty Liver Disease (NAFLD)/Non-Alcoholic Steatohepatitis (NASH) with Sex Steroids

Non-alcoholic fatty liver disease (NAFLD) includes the entire spectrum of steatohepatitis as a non-alcoholic steatohepatitis (NASH) with or without fibrosis, cirrhosis and hepatocellular carcinoma (HCC) [52,53,54,55], related to systemic features [56,57] and excessive mortality from cardiovascular and liver diseases [58,59,60,61]. Histologically indistinguishable from alcoholic liver disease, the NAFLD [62] is closely related to insulin resistance [63] and metabolic syndrome [53,54].

There are studies which show that androgens protect against NAFLD [43], because low serum T levels and hepatic steatosis in men are closely related [64]. However, other reports show opposite results, with androgens promoting the development and progression of NAFLD [65,66]. In vitro data similarly suggests that exposure to excessive amounts of androgens (including corticosterone) can lead to lipogenesis [12]. These inconsistencies may result from the use of various animal models, genders, methods of treatment or combinations of various steroid hormone replacements. In addition, it is the T to DHT ratio that is most important for the development and progression of NAFLD rather than the concentrations of T or DHT [67]. In the human liver, both isoforms of 5α-reductase (5αR1, 5αR2) are present, and the level of the first isoform becomes higher with the growing severity of NAFLD symptoms. Mice with 5α-reductase knockout (5αR1^−/−^, 5αR2^−/−^) do not convert testosterone into DHT. Implementing an ALIOS diet for these knockout mice induced a development of great hepatic steatosis only in 5αR1^−/−^, but not 5αR2^−/−^ [12]. This steatosis was driven largely by impaired corticosterone clearance rather than decreased DHT [12]. Similarly, male 5αR1-knockout mice on a HFD diet also demonstrated hepatic steatosis as a result of hepatic reduction in FA β-oxidation and increased TG accumulation [13]. The authors of the mentioned study also observed hepatic steatosis in obese male Zucker rats, both intact and castrated, after treatment with finasteride (5α-reductase type 2 inhibitor) [13]. The hepatic steatosis was independent of DHT, but changes in 5αR1 activity with non-selective 5α-reductase inhibition in overweight men with prostate disease could indicate the beginning and progression of hepatic metabolic failure [13].

In another study [68], a very low T serum level in feminised (Tfm) male mice on a normal diet showed increased lipid accumulation although this was significantly less than cholesterol-fed Tfm mice. Tfm mice on a normal diet demonstrated increased gene expression of hormone sensitive lipase, stearoyl-CoA desaturase-1 (SCD1) and PPAR-γ, although acetyl-CoA carboxylase alpha (ACACA) and FASN were not altered. Yet testosterone supplementation caused a reduction in the lipid deposition in the liver of Tfm mice compared to placebo-treated Tfm as a result of a decrease in the expression of key regulatory enzymes of fatty acid synthesis [68]. Hepatic AR-knockout (H-AR^−/y^) male (but not female) mice on a HFD diet also developed hepatic steatosis as a result of a rise in SREBP-1c and PPAR-γ [4,11]. Moreover, the insulin resistance of these male mice was associated with a decline of phosphoinositide-3 kinase (PI3K) action, increased phosphoenolpyruvate carboxykinase (PEPCK) expression, and correlated with increased protein-tyrosine phosphatase 1B expression (PTP1B). Loss of AR in aging H-*AR*^−/*y*^ male mice caused a rise in hepatic TG volume, so that hepatic androgen receptors may be a key for avoiding liver steatosis development. Lin et al. [11] proposed the development of AR agonists to target hepatic AR and thus improved the effectiveness of therapies used in metabolic syndrome in male patients. Male mice with complete (not only hepatic) AR knockout (ARKO) developed increasing triglyceride deposition in liver, obesity, and severe IR [4]. As hepatic AR has a greater effect in men than in women [7,11], Kanaya et al. [69] performed an experiment to try to better understand how elevated androgen levels regulate food intake and obesity in females. Ovariectomized female mice treated with DHT (non-aromatazable androgen) exhibited increased food intake, significantly higher lipids storage in the liver, and other signs of biochemical dysfunction (increased fasting glucose, impaired glucose tolerance, resistance to leptin) [69].

The aforementioned reports indicate that androgens have a major influence on lipid metabolism in female livers. There are also many indications that hyperandrogenic women with PCOS may indirectly increase the risk of NAFLD by obesity, IR, and directly by the hepatotoxic effect (significantly increased level of alanine aminotransferase (ALT)) [70]. Compared to premenopausal women, men and postmenopausal women have higher LDL-C and lower HDL-C concentration in blood, so estrogens could play an important role in decreased hepatic fats storage [71]. This indicates that menopause is related to increased body weight and higher risk of metabolic diseases. In an OVX mice model of menopause [72], increased adiposity was prevented by estrogen replacement. In that study, treatment with E2 was associated with general reduction of adipose tissue mass (because of down-regulation of lipogenic genes under the control of SREBP-1c). In the liver, endogenous E2, similar to the adipose tissue, caused a decrease in the expression of lipogenic genes. It was shown by D’Eon et al. [72], that in the liver, estradiol participated in free fatty acids dividing during oxidation and prevented TG storage by up-regulation of *PPAR-δ* and by direct initiation of AMP-activated protein kinase (AMPK). Accordingly, genomic and non-genomic actions of E2 promote leanness in OVX mice independently of reduced energy intake [72].

There is ample evidence from screening studies that the prevalence of NAFLD is higher in males compared to females, regardless of age [73,74,75,76,77,78]. In a study examining the incidence of NAFLD in women, 7.5% of those going through menopause and 6.1% of postmenopausal women were found to have NAFLD, in comparison to 3.5% of premenopausal women [76]. The increased risk of NAFLD did not correlate with hormone replacement therapy. The incidence of NAFLD in women rose with age, but did not change with age in men [76]. Thus, this indirectly indicates that endogenous (contrary to exogenous) estrogens could play a protective function against the advancement of NAFLD in women. On the other hand, there is data indicating that hormone-replacement therapy may lessen the risk of diabetes, but its mechanisms are unclear [79]. In contrast, an Italian multicentre study on almost 5500 healthy hysterectomised women who received tamoxifen or placebo for five years showed that the medicament increased the risk of NAFLD/NASH development only in overweight and obese women with features of MetS [80]. A study on women with T2D documented that low doses of hormone replacement therapy significantly reduced liver enzymes: AST, ALT, GGT (γ-glutamyltransferase), and ALP (alkaline phosphatase) in serum, potentially due to a reduced level of hepatic fat accumulation [81]. Authors of this publication indicate that the explanation for the HRT improvement of liver physiology could help in the search of the effective treatment of non-alcoholic fatty liver disease among women. (Summary of the above data in Table 2.)

## 3. HBV/HCV and Hepatocellular Carcinoma (HCC, Hepatoma)

Hepatitis B (HBV) and hepatitis C (HCV) are two hepatotropic viruses belonging to the family of Hepadnaviridae and Flaviviridae (respectively), differing in genome, life cycle and molecular prediction. HBV is a DNA virus that has an ability to integrate into the DNA of the host cell. In contrast, HCV is an RNA virus that replicates in cytoplasmic membranous host cell networks. The innate and adaptive immune responses are the main mechanism involved in determining persistent hepatitis infection, and the innate immune response is the first line of defence against viral infections [82].

The HBV contagion and subsequent consequences of infection are different in males and females [83,84,85,86,87,88,89]. The effects of sex differences, especially sex hormones, on the innate immune response to HBV are largely unknown, which is at least partly due to the lack of appropriate research models. Slightly more is known about gender differences in the adaptive immune response to HBV infection. For instance, after a prophylactic vaccination against HBV, women have a higher titre of anti-HBV antibodies than men [90]. Hepatocellular carcinoma (HCC) development, pathogenesis and disease progression-induced hepatitis B infection show gender-related differences [91]. HBV-related HCC occurs more often in men than in women [92]. Rates of liver cancer in men are typically 2 to 4 [93] or even 3 to 5 [94] times higher than in women. Gender-related variation in liver cancers is common in mammals, from rodents to humans, and was firstly described in mice in the late 1930s, with female mice being resistant to liver cancer [95]. The remarkable gender disparity suggests an important role of sex hormones in HCC pathogenesis [96]. It is probable that the specific immune response of the host is reflected in HBV replication and viral protein levels. Likewise, in a study conducted on HBV infected mice, males had up-expressed DNA and protein of HBV in comparison to females. The reduced functionality (not the number) of CD8^+^ T lymphocytes was accompanied by increased numbers of regulatory T cells (T reg) in males which may explain why, among male HBV human patients, there are more infections and more failed cases of immunotherapy than in women [91].

### 3.1. Relationship between Estrogens/ERs and HBV-Related Acute Liver Failure Such as HCC

The sexual dimorphism of hepatitis B virus-related liver diseases may be related to estrogen and its receptors. One possible explanation is that the ERα polymorphism leads to a defective immune cell response to estrogen in HBV-related acute liver failure [97]. Antiviral modulation of immune responses by sex hormones can also help to explain the prevalence of HCC in men, as in the case of chemically induced HCC by diethylnitrosamine (DEN, a chemical carcinogen), which is more severe in males than in female mice, due to an increased production of IL-6 by Kupffer cells (in a manner dependent on the Toll-like receptor adaptor protein MyD88) in the male liver [98]. Interleukin 6 (IL-6) is a multifunctional cytokine that is largely responsible for the hepatic response to systemic infection or inflammation, often referred to as an ‘acute phase response’ [99]. Naugler et al. [98] demonstrated that estrogens inhibited IL-6 by reducing the activation by Myd 88-induced NF-κB. Physiological doses of estrogens can suppress metastasis of HCC not only by decreasing IL-6 expression but also by decreasing hepatocyte growth factor levels [100]. The hepatocarcinogenic effect of IL-6 in hepatocytes can be stopped by inhibiting transcription factor STAT3 and reducing the activity of mitogen-activated protein kinase JNK (c-Jun N-terminal kinases) [98]. The protection against the development of liver cancer in carcinogen-treated mice also depends on ERα-mediated estrogen signaling of forkhead box protein A (Foxa) factors such as Foxa1 and Foxa2 [101] pioneer transcription factors in the liver, crucial for steroid hormone signalling (estrogens and androgens) as essential controllers of variations of liver cancer in terms of gender [95]. The integrative genomic analysis showed that the risk of HCC in women might be associated with the SERPINA6-rs1998056 regulated by FOXA/ERα [102].

### 3.2. Relationship between Androgens/AR and HBV-Related Acute Liver Failure Such as HCC

Mechanisms through AR that can mediate the expansion of HCC also include the modulation of innate immunity. Shi et al. [103]. showed that AR could suppress IL-12A expression at the transcriptional level via direct binding to the IL-12A promoter region which results in repressing the efficacy of natural killer (NK; related innate immune surveillance) cell cytotoxicity against liver cancer cells. On the other hand, there is also evidence that activated AR can inhibit HCC metastasis by inducing cellular apoptosis by modulation of p38 kinase phosphorylation [104], shown to be mitogenic-dependent and playing a significant role in HCC [105,106,107].

In addition to affecting the immune response, sex hormones can also directly affect the activity of the virus. In general, the HBV surface antigen (HBsAg) circulates at a higher level in the serum of male mice than in females [108], and its level decreases after castration, thus indicating that the expression of the viral antigen and viral replication are regulated by androgens [109]. The HBV genome integrated into the host cell DNA contains two androgen respond elements (ARE) in the enhancer region I. When the AR-androgen complex is internalized to hepatocytes, it binds to both the nuclear and viral ARE sequences, thereby activating the transcription of the HBV genome and the production of HBV X (HBx) protein [110]. This protein, in turn, facilitates dimerization of AR and enhances transcriptional activity of AR by activating Src kinase, thus creating a feedback loop that can promote hepatocarcinogenesis [92]. The AR further acts in conjunction with other molecules, such as cell cycle-related kinases (CCRKs), which in turn activate oncogenic β-catenin in hepatocytes. This mechanism indicates that androgens/AR signalling may promote the development of HBV-related hepatocellular carcinoma and explains the higher incidences of HCC as well as higher HBV titres in male serum than female [111]. Conversely, estrogen signalling probably inhibits hepatocarcinogenesis and protects against HBV-related HCC progression. The molecular mechanism of estrogen is mediated by the binding to the nuclear ERα which inhibits the enhancer I of HBV and transcription of integrated viral genomes [92,111].

### 3.3. Complicity of Noncoding mRNAs in the Onset and Progression of HCC

Progression of HCC is also related with several long noncoding RNAs such as lncRNAs, which have miR-374b/421 and miR-545/374a clusters. Considering that the estrogen-related receptor gamma (ESRRG) is a potential target gene of miR-545, it has been hypothesized that this mechanism may be associated with a lower incidence of HVC-induced HCC in women. As the miR-545 and miR-374a were up-expressed in male *versus* female HCC individuals in a study by Zhao et al. [112], the authors of the study concluded that the up-expressed miR-545/374a cluster could be related to a low chance of recovery, and suggested the employment of miR-545/374a levels in sera for HCC diagnostics. The role of E2 in regulating the activation of p53 and miR-23a expression could be crucial to understanding the sex differences observed in HCC [113]. In miRNA PCR array, Huang et al. [113] found more than a two-fold alteration in apoptotic miRNAs (25 was upregulated and 10 was downregulated) in E2-treated cells. Among these miRNAs, expression of miR-23a was related to p53 functional status in the male-derived liver cell-lines. Moreover, miR-23a expression correlated negatively with the expression of target gene X-linked inhibitor of apoptosis protein (XIAP), but positively with the caspase-3/7 activity. So, a decrease of XIAP may contribute to caspase-3 activity and cell apoptosis. The authors of the study emphasize the huge potential of miRNAs as biomarkers and therapeutic agents thanks to their ability to control gene expression. In research in which lentivirus-mediated ERα small interfering RNA (siRNA) was transfected into HCC cells (Hep3B), the downregulation of ERα expression caused the inhibition of cell proliferation, reduced cell invasion, slowed-down cell population at S phase, and increased the rate of apoptosis [114]. According to these authors, ERα may play a very important role in carcinogenesis of HCC and its knockdown may offer a new potential gene therapy approach for human liver cancer in the future. In addition, it has been proved that the promotor of pri-miR-216a has an androgen-responsive element [115]. The up-expression of miR-216a was mainly noticed in male patients, as a result of transcriptional activation of pri-miR-216a by the androgen signaling further reinforced by X protein (HBV protein) [115].

### 3.4. HCC Malignancy and Sex Hormones

Generally, a correlation between the axis of androgen/androgen receptor and HCC incidence have been confirmed, but the mechanism is still largely unknown. For example, it is proposed that androgen/AR complex after binding to promoter of Nanog (pluripotency factor) can promote HCC stemness. It is worth emphasizing that, in HCC tissues, AR expression was abnormally high and showed a correlation with Nanog expression [116]. Another study revealed a “vicious circle” of androgenic signaling. This signaling increases the expression of CCRK (cycle-related kinase, a direct AR transcriptional target), which results in the activation of the Wnt/β-catenin/TCF (T cell factor) pathway that finally leads to up-expression of AR in HCC cells [117]. CCRK was overexpressed in approximately 70% of HCCs and was significantly correlated with tumor staging. Thus, the interaction of AR/CCRK stimulates cell cycle progression and induces tumor formation (promotion of hepatocarcinogenesis) [117]. It was noted that the expression of matrix metalloproteinase 9 (MMP9), an important marker of migration, adhesion, invasion and metastasis of liver cancer [118], was higher in HCC tumors in mice lacking specific AR in the liver (L-AR^−/y^) compared to WT-animals. It was also found that AR suppresses cell migration via suppression of nuclear factor kappa B (NF-κB)-MMP9 pathway [104]. In their next paper, the authors showed that AR affects cell adhesion and cellular mobility through the AR-β1-integrin-PI3K/AKT signaling pathway in HCC [119]. The L-AR^-/y^ mice with carcinogen-induced HCC developed more undifferentiated and larger size tumors at the metastatic stage and died earlier with increased lung metastasis [104]. These results indicated that hepatic AR may play dual opposite roles, to promote HCC initiation but suppress HCC metastasis.

DEN-injected female mice exhibited scarcer dysplastic foci and less acute early stage of HCC than males, with more differentiated tumors and fewer metastases [120]. Castration of these mice down-regulated cyclin E kinase and amplified hepatocyte apoptosis, and estradiol/progesterone enhanced those effects. In control female mice, cyclin E kinase activity was lower than in males, and testosterone administration of ovariectomized mice increased cyclin E and its kinase activity and accelerated hepatocarcinogenesis. Moreover, exogenous testosterone not only up-expressed cell cycle regulators (cyclin D1 and E, CDK2) but also down-expressed p53 and p21, which improved hepatocyte viability. Conversely, E2 inhibited hepatocyte cell cycle markers, increased p53 and reduced hepatocyte and HCC viability. This study showed that both sex hormones determined the male predominance to hepatocarcinogenesis: castration of male mice delayed the onset of HCC [120]. Moreover, the DHT to T ratio is also an essential indicator, because it is elevated in patients with HCC in contrast to patients with cirrhosis or healthy individuals [121]. Furthermore, the size and cell division activity of HCC significantly declines after blood DHT levels drop [122]. Therefore, in terms of tumorigenesis, DHT (a more active T metabolite and AR ligand) cannot be omitted. According to Yu et al. DHT reinforces hepatocellular carcinoma cell division depending on AR activation [123], and the decline in HCC malignancy after AR antagonism treatment is linked with a decrease in blood DHT [124]. This data confirms the observations of Dowman et al. [12], where more than half of the mice after one year of the ALIOS diet revealed hepatocellular lesions similar to those observed in HCC, compared to one-fifth of 5αR2^−/−^ and zero of 5αR1^−/−^ (isoform 5α-reductase knockout) mice. Because of this, it has been proposed that the 5αR1 deletion could have protective function against the NAFLD-associated HCC expansion, and this enzyme isoform may become a therapeutic target [12].

### 3.5. Hypothesis about the Role of ERs in HCC

Although hepatocellular carcinoma is known to be accompanied by decreased expression of ERs, their role in HCC is not fully understood [125]. There are some studies on the effects of estrogen/ERs signaling on various tumor suppressors, but their results are inconclusive. The development and invasion/progression of HCC and other cancers are associated with metastasis-associated protein 1 (MTA1) gene expression [126,127,128]. Additionally, the results of research carried out by Deng et al. [129] show that ERα up-regulation inhibits the division and spread of HCC. On the other hand, the MTA1 overexpression lowers ERα-controlled inhibition of HCC cells’ division and metastasis. These results indicate a co-regulation of ERα and MTA1 in the response to HCC, providing a basis for understanding the gender-related difference in HCC progression. Overexpression of ERα has also been shown to mediate apoptosis in ERα-negative Hep3B cells via the binding of ERα to specificity protein 1 (SP1). Then this complex (ERα-SP1) binds to the TNFα gene promoter, inducing the expression of active caspase 3 in a ligand-dependent manner [125]. It was also shown that decreased expression of ERα mRNA due to inhibition of ion channel (KCNN4; Ca^2+^-activated K^+^ channel) by TRAM-34 (1-[(2-chlorophenyl)diphenylmethyl]-pyrazole) led to a decrease in activation of NF-kappaB, the factor known to be involved in the development of HCC [130]. Therefore, TRAM-34 is proposed as a new therapeutic target for the treatment of HCC.

In addition, E2 may also inhibit the progression of HCC, since E2-suppressed cell cycle markers, increased p53-regulated p21, Bcl-X_L_ and Bax expression, consequently reducing the viability of HCC cells [120]. Interestingly, estradiol was shown to have a dual effect: in hepatocytes, increasing estradiol concentrations promoted cell survival, while the opposite effect was observed in HCC cells. A primary culture of hepatocytes and HCC cells clearly responded differently to estradiol stimulation with respect to cell death [120]. These dual effects of estradiol have been described before: low doses of endogenous estradiol are tumor-enhancing, while high doses of exogenously delivered estradiol inhibit tumor formation [131,132]. This is probably why some of the results of research on the effects of estrogen on HCC are contradictory, in addition to the existence of various estrogen receptor variants [133]. One of ER’s alpha receptors is vER (variant estrogen receptor) which does not have the hormone-binding domain but has a normal DNA-binding domain, responsible for the transcription of estrogen-dependent genes [134]. In chronic hepatitis, vER transcripts, in contrast to wtER (wild-type ER), are present more frequently in men and in HBsAg-positive subjects than in individuals with antibodies to HCV. In HCC male patients the vER transcript is overexpressed or is the only one expressed form [133]. The much more frequent presence of vER in men, mainly those with HBsAg, both in the early stages of the disease and chronic hepatitis, indicates that this variant of estrogen receptor promotes the uncontrolled proliferation and development of hyperplasia, and may be a mechanism of neoplastic alteration in men [133]. The hepatocellular carcinoma cells that express vER are highly malignant [134,135], because of the raised proliferation rate and because they are insensitive to tamoxifen (antiestrogen). Fortunately, megestrol (a drug that blocks wtER and vER) does have some influence on the success of therapy in HCC with the expression of vER [134].

### 3.6. Immune Response in Liver Failure and Sex Hormones

Yet another role of estrogen receptors in HCC progression was shown by Wei et al. [136] who presented a novel link between estrogen receptor β and the NLRP3 inflammasome (an intracellular multiprotein complex involved in the innate immune response to pathogens) in hepatocarcinogenesis. They demonstrated that expression of ERβ was significantly downregulated in HCC tissue compared with normal liver tissue; moreover, ERβ expression had a significant negative correlation with disease progression and a positive correlation with the expression level of NLRP3 inflammasome components. It is known that loss of NLRP3 inflammasome in HCC tissue contributes to tumor progression. Treatment with 17β estradiol significantly inhibited the malignant behavior of HCC cells through E2/ERβ/MAPK pathway-mediated upregulation of the NLRP3 inflammasome [136]. E2 could achieve the same effect (suppression of tumor growth) via regulating the polarization (producing distinct functional phenotypes as a reaction to specific microenvironmental stimuli/signals) of macrophages [137]. During this process, 17β-estradiol suppressed macrophage activation and HCC development alternatively by inhibiting the interaction between ERβ and ATPase coupling factor 6 (ATP5J, an ATPase component), and then blocking the JAK1-STAT6 signaling pathway [137]. These results could contribute to the implementation of a new HCC therapeutic strategy based on the discovered aforementioned mechanism.

### 3.7. Activity of Aromatase/Estrogens/ERs (and Variants) in HCC

A study by Carruba et al. [138,139] carried out on nontumoral, cirrhotic, and malignant human liver tissue samples (in vivo) and in HepG2, HuH7, and HA22T cells (in vitro) revealed for the first time that the level of the aromatase enzyme is significantly increased in liver cancer cells (malignant human liver tissue and HepG2 hepatoma cells), which leads to an increase in the local conversion of estrogens from androgens. Aromatase expression is moderate (or intermediate) in cirrhotic human liver samples (or HuH7 cells) and undetectable (or very low) in nontumoral human liver tissue (or HA22T cells) [138,139,140]. The level of local androgen aromatization is correlated with the degree of malignancy of the liver tissue/cell line. Therefore, locally elevated estrogen formation affects the development and progression of cancer tissues and cells (HCC, HepG2) by activating the rapid signalling pathway mediated via amphiregulin (AREG; a member of the EGF family), a ligand of EGF-R (epidermal growth factor receptor) [138,140]. Moreover, elevated expression of AREG corresponds with ubiquitous expression of hERalpha46 (human variant of ERα) [138,139] and occasional expression of the hERβ2/Cx (human variant of ERβ) [139]. Either none or a low expression of wild-type ERα and ERβ is observed in liver cancer cells and malignant tissues, and the pattern of wtERα is inversely related to aromatase expression [140]. Therefore, the elevated estrogen production induced to a higher aromatase activity could induce liver tumor cell growth through a variant ERα-mediated mechanism. Furthermore, the modification in activity of aromatase-estrogen-amphiregulin-EGF-R axis in issue injury or inflammation could result in growth of tumours such as liver, breast or prostate and progress of chronic diseases such as diabetes, obesity, Alzheimer’s and heart disease [140]. Other studies also confirm that the change in ERα status (from wild type ERα66 to the ERα36 splice variant, but not to the ERα46 splice variant) influences HCC development [141]. Probably, due to the existence of numerous ER splicing variants with diverse action, many HCC patients have not responded properly to anti-estrogen treatment. This was possibly caused by hERα66 which inhibits the activation of hERβ in an estrogen-dependent and independent manner [142].

Both prognostic factors and survival rate after therapeutic HCC resection differed between sexes, with female patients having a better overall survival rate than male patients (women had a less invasive tumor phenotype), but this survival benefit was only observed in cases of tumor-node-metastasis stage I diseases compared with males at the same stage; although female patients had a greater prevalence of increased serum alpha-fetoprotein (AFP), AFP and tumor number had prognostic significance only for males; vascular invasion and serum g-glutamyl transpeptidase (GGT) levels were independent risk factors for early recurrence in female patients, whereas AFP and GGT level were independent risk factors for late recurrence [143]. These authors suggest that because estrogens may have a protective effect against early-, but not late-stage, HCC, more aggressive treatment should be attempted for female patients with recurrent HCC [143]. The effective treatments for hepatocellular carcinoma are hepatectomy and liver transplantation, although the risk of recurrence is still high, particularly in patients with a large pool of circulating cancer cells (CTCs) positive for cancer stem cell/progenitor cell mercers. In this area, the results of a study performed on two AR knockout mouse models with spontaneous HCC, which showed a negative relation between HCC recurrence/progression after hepatectomy expression of AR in CTCs, are very interesting and promising. AR-regulated suppression of HCC is a solid sign that this receptor could act as a gatekeeper of HCC recrudescence after surgery [144].

## 4. Other Pathological Conditions and HCC

It was also shown that cirrhosis, as a result of liver fibrosis in chronic liver disease (CLD), could lead to neoplasia in hepatocellular carcinoma [145]. A cohort study of over 12,000 patients showed that males with CLD were younger (52.9 vs. 58.7 yrs.) and additionally more frequently suffered from alcoholic liver disease (11.4% vs. 6.9%) than women with CLD [146]. Researchers of this analysis have highlighted significant gender differences in terms of the etiologic factors and the onset of chronic liver disease. On this basis, it can be concluded that fibrosis as a consequence of CLD may also be gender-dependent. According to Saginelli et al. [146] factors such as NF-κB, STAT3 and JNK could be linkers with the onset of HCC in patients with cirrhosis. In an inflammatory mouse model (mdr2^−/−^ mice with cholangitis, chronic liver inflammation and finally HCC), the TNF-NF-κB axis had a pro-carcinogenic effect on the liver. It was demonstrated that inhibition of NF-κB by doses of anti-TNF-α stopped HCC progression [147]. In lymphotoxin (LT) transgenic mice models, the overexpression of LT is related with chronic inflammation and infiltration into the liver by T, B and dendritic cells, with cytokine (IL-1β, IFN-γ, IL-6) over production reaching the highest concentration in HCC. These mice also had elevated production of chemokines (CXCL1, CCL7, CXCL10) as a result of NF-κB activation [148]. Activation of NF-κB is a frequent and early event of human HCC [149,150]. As it had been described earlier [98,100], by reducing Myd 88-induced NF-κB or STAT3/JNK kinase-pathway, estrogens may inhibit IL-6 or hepatocyte growth factor, and then this activity can suppress the progression of liver fibrosis and chronic liver disease. It has also been noted that there are noncirrhotic patients with HCC that have a better overall survival and disease-free survival than cirrhotic patients with HCC [145].

A not often noted and not well known progression is the development of hepatocellular carcinoma in patients with primary biliary cirrhosis (PBC). An early study carried out on almost 400 patients with PBC has shown that only 14 patients developed HCC; and the appearance rate was higher in patients with advanced-stage PBC, with age at the time of diagnosis and male gender more associated with the development of HCC [151]. In the decade following, there was the point of view that the disease overwhelmingly affected females. In epidemiological studies, only 7–11% of documented PBC patients were males, but with a higher risk of life-threatening complications such as gastrointestinal bleeding and hepatoma [152]. In PBC, females demonstrated enhanced antibody production and cell-mediated responses, in addition to an increased CD4 T cell number, probably because, normally T decreases IgG and IgM production by plasmocytes in healthy males and females, or because of co-expression of ER and AR on B cells, whereas CD8 T cells, monocytes, neutrophils and NK cells express only ER [152].

Obesity related to leptin secretion is also a significant predictor of HCC in humans. In the context of sex steroid dimorphism, it is not known whether estrogens antagonize the action of leptin in women. HCC line HepG2 cells cultured with leptin and E2, PPT, DPN (bis-hydroxy-phenyl-propionitrile, a ERβ selective agonist) or G-1 (GPER selective agonist) were studied. The results of the experiment showed that E2/ERs upset the oncogenic function of leptin in the HepG2 cells via preventing their division and promoting their death; and these events were linked with regression of changes in SOCS3/STAT3 induced by leptin, up-regulation of p38/MAPK as a result of ERβ action, and up-regulation of ERK due to the action of ERα and GPER. Additionally, it was shown that agonists of ERα, ERβ and GPER induced cell apoptosis in the HepG2 line [153]. This further data demonstrates the protective role of estrogens in the expansion of HCC, and that estrogen receptors could be a target in the prevention/treatment of leptin-induced HCC.

## 5. Pathological Condition Associated with the Biliary Tree

The biliary tree (network of intra- and extra-hepatic bile ducts) is lined with a specific type of epithelial cell known as cholangiocyte. They are a heterogeneous (biochemically and functionally) highly dynamic population of cells that modify (via transcytotic transport of various ions like Cl^−^, HCO_3_^−^, Ca^2+^, Na^+^, K^+^, solutes, water and also glucose) hepatocyte-derived bile under the direction of hormones, cytokines, growth factors and neurotransmitters [154]. Other functions of cholangiocytes are proliferation, injury repair, fibrosis, angiogenesis and regulation of blood flow [155]. It was documented that cholangiocytes can undergo damage or pathological proliferation during chronic cholestatic liver diseases (cholangiopathies), primary biliary cirrhosis (PBC), primary sclerosing cholangitis (PSC), polycystic liver disease (PCLD) and cholangiocarcinoma (CCA) [155]. During biliary fibrosis, proliferating bile duct epithelial cells, along with hepatic stellate (Ito) cells, are the dominant source of connective tissue growth factor (CTGF); additionally, in this pathological condition, the elevated mRNA level of TGF-β1 that is produced not only by Ito cells but also by activated cholangiocytes seemed to be the main source of this profibrogenic factor [156]. Within the hepatic parenchyma are also oval cells (stem cells). These cells are heterogeneous and bipotent in terms of their developmental maturity or their commitment to either the hepatocytic or biliary lineage. Studies in rodents demonstrated that oval cells not only are associated with an increased risk of HCC in chronic liver disease, but also can proliferate and form ductule-like structures during carcinogenesis and biliary obstruction, and have been also indicated to have the potential involvement of bile epithelium in fibrosis associated with other chronic liver diseases [157].

Suggestions exist that hormones, especially the sex hormones, play a key role in the modulation of cholangiocyte growth in a damaged liver [155]. For example, bile duct ligation (BDL) caused an increase in ER-β expression in cholangiocytes in comparison to control animals [158]. Clinical studies have shown that patients with late-stage PBC had markedly reduced ER expression in cholangiocytes. ER modulators improve the serum parameters of cholestasis in PBC patients [159]. An in vitro study documented that estrogens, by Src-Shc-ERK1/2 signalling mechanisms, modulate cholangiocyte proliferation and secretion [160,161]. This was confirmed in an experiment on rats with bile duct ligation after tamoxifen or ICI 182,780 9 (anti-estrogen) treatment; the BDL rats had significantly lower weight of intrahepatic bile ducts (IBDM) compared to the control as a result of impaired proliferation and increased apoptosis [155]. Another experimental cholestasis study showed that ovariectomised (OVX) female rats after BDL had significantly reduced bile duct mass associated with a decreased expression of ERβ. Exogenous E2 caused a normalization of bile duct mass, ERβ expression and cholangiocyte proliferation in comparison to untreated BDL rats [162]. This is why it is highly likely that estrogens might delay the progress of cholangiopathies into ductopaenia [163].

Generally, there is little data that described the influence of androgen on biliary epithelium. For example, Yang et al. [164] showed the expression of AR in cholangiocytes and that testosterone stimulated biliary growth and secretion during cholestasis. The cAMP level in cholangiocytes from BDL rats was higher than cAMP levels from normal cholangiocytes [165,166]. Castration of the BDL rats inhibited the stimulatory effects of secretin on cAMP levels in cholangiocytes, and bile and bicarbonate secretion in bile fistula rats; and exogenous testosterone restored the functional secretory activity (secretin stimuli bile and bicarbonate secretion) of cholangiocytes in the castrated BDL rats [164]. Reduced serum testosterone levels as a result of castration or anti-testosterone treatment led to a decrease of IBDM in normal and BDL rats in comparison to the non-castrated rats; and then endogenous testosterone partly compensated for the castration-induced loss of IBDM. Moreover, in the bile duct of BDL castrated rats and BDL rats treated by anti-testosterone, there was an increase in apoptosis compared with BDL rats [164].

On the basis of the above mentioned studies, it was proposed that not only estrogens [155] but also testosterone is important in sustaining biliary proliferation and ductal secretory activity in pathological conditions like functional damage of the biliary epithelium [164]. 

## Figures and Tables

**Figure 1 ijerph-17-02620-f001:**
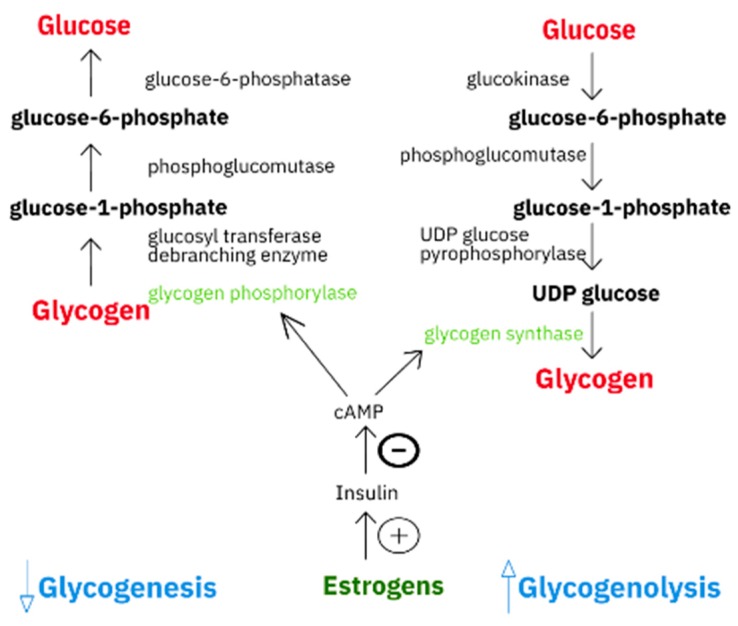
Graphical comparison of glycogenesis and glycogenolysis pathways.

**Table 1 ijerph-17-02620-t001:** Summary data on impact of the hormone imbalance, disturbance of the sex hormone receptors and the enzyme activity/expression on the hepatic metabolism of carbohydrate in relation to the gender as a cause of many physiological dysfunctions, syndromes or diseases.

Hormone Imbalance or Receptor/Enzyme Dysfunction	Results	References
**Disturbed carbohydrate metabolism in male**		
↓ Testosterone	Hyperglycemia, T2D, MetS	Muthusamy et al. [5]
AR (lack)	IR, T2D	Lin et al. [4]
AR knockout	↓ Glucose metabolism, IR, hyperinsulinemia	Lin et al. [4]
AR knockout + HFD	↓ Sensitivity to insulin	Lin et al. [11]
5α-red1 knockout + ALIOS	Hyperinsulinemia	Dowman et al. [12]
5α-red1 knockout + HFD	Hyperinsulinemia	Livingstone et al. [13]
↓ ERα	↓ Glucose metabolism, IR, hyperinsulinemia, T2D, MetS	Zirilli et al. [16], Yamada et al. [28]
ERαKO	↓ Glucose tolerance, hepatic IR	Bryzgalova et al. [20], Nilsson et al. [29]
↓ Testosterone	Hyperglycemia, T2D, MetS	Muthusamy et al. [5]
**Disturbed carbohydrate metabolism in female**		
↓ Testosterone	↑ Glucose	Kelly et al. [7]
↓ Estrogens	↑ Diabetes	Saengsirisuwan et al. [21], Feigh et al. [22]

ALIOS—American lifestyle-induced obesity syndrome, AR—androgen receptor; ERα—estrogen receptor alpha, ERαKO—ERα knockout, HFD—high-fat diet, IR—insulin resistance, MetS—metabolic syndrome, T2D—type 2 diabetes, 5α-red1—5α reductase type 1.

**Table 2 ijerph-17-02620-t002:** Summary data on impact of the hormone imbalance, disturbance of the sex hormone receptors and the enzyme activity/expression on the hepatic metabolism of lipids in relation to the gender as a cause of many physiological dysfunction, syndromes and diseases.

Hormone Imbalance or Receptor/Enzyme Dysfunction	Results	References
**disturbed lipids metabolism in male**		
↓ Testosterone	↑ Hepatic steatosis ↓ Hepatic steatosis	Vőlzke et al. [64] Jones et al. [65] Schwingel et al. [66]
5α-red1 knockout + ALIOS	Hepatic steatosis	Dowman et al. [12]
5α-red1 knockout + HFD	↑ TG, hepatic steatosis	Livingstone et al. [13]
Hepatic AR- knockout + HFD	Hepatic steatosis, IR	Lin et al. [4], Lin et al. [11]
↓ AR	↑ TG in liver, hepatic steatosis	Lin et al. [4]
ARKO	↑ TG in liver, obesity, IR	Lin et al. [4]
ArKO	Liver steatosis, obesity	Hewitt et al. [40], Fisher et al. [44]
↓ Aromatase	Metabolic anomalies	Maffei et al. [15]
↓ ERα + HFD	↑ TG, ↑ diacylglyceride, IR	Zhu et al. [38]
ERαKO	Fatty liver	Bryzgalova et al. [20]
LKO + ↓ Gpr30 + HFD	↑ TG, ↑ diacylglyceride	Zhu et al. [38], Meoli et al. [39]
**disturbed lipids metabolism in female**		
↓ Estrogen	↑ LDL-C, ↓ HDL-C, hepatic steatosis	Trapani et al. [71], Chen et al. [1]
↓ Estrogen + HFD/HFHC	NASH	Kamada et al. [47]
LKO + ↓ Gpr30 + HFD	↑ TG, ↑ diacylglyceride, ↓ HDL-C, ↑ fat liver accumulation, IR	Zhu et al. [38], Meoli et al. [39]
ERαKO	Fatty liver	Bryzgalova et al. [34]
ArKO	Liver steatosis, obesity	Fisher et al. [44]
Hyperandrogenism + PCOS	Obesity, IR, NAFLD	Bohdanowicz-Pawlak et al. [70]

ALIOS—American lifestyle-induced obesity syndrome, AR—androgen receptor, ARKO—AR knockout, ArKO—aromatase-knockout, ERα—estrogen receptor alpha, ERαKO—ERα knockout, Gpr30—orphan G protein-coupled receptor 30, HDL-C—high-density lipoprotein HFD—high-fat diet, HFHC—high-fat an high-cholesterol diet, IR—insulin resistance, LDL-C—low-density lipoprotein, LKO—liver ERα knockout, MetS—metabolic syndrome, NAFLD—non-alcoholic fatty liver disease, PCOS—polycystic ovary syndrome, T2D—type 2 diabetes, TG—triglycerides, 5α-red1—5α reductase type 1.

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
