# Peer review of "Sex Hormone-Dependent Physiology and Diseases of Liver"

_ijerph, 2020, doi:10.3390/ijerph17082620_

Round 1

Reviewer 1 Report

In review submitted authors described how sexual dimorphism play a role in several fields including liver metabolism. Moreover, they indicated that difference in sex hormone levels and sex hormone-specific gene expression could modify the incidence of gastrointestinal diseases, in particular liver diseases.

The review is well done; however I would like to suggest some minor revision in order to increase the reading.

  • Each paragraph is too long, and it could be hard following the scheme logic all time. For this reason, I would like to suggest to authors to divide every paragraph in shorter subparagraph indicating the specific point described.
  • Authors claim HBV and HCV as some risk factors for HCC. However, other pathologies such as cirrhosis could induce HCC into liver. Authors should add a briefly discussion about other pathologies.
  • The difference in liver metabolism influenced the biliary tree metabolism and this could play a role in onset biliary diseases. Could the authors add a discussion about liver-tree metabolic link in this review?

Author Response

Dear Reviewer,

Thank you for the work put in Review of mine manuscript. I would also like to thank you for your valuable comments and instructions. I have improved my manuscript and every changes have been marked in blue (or green – see lower).

  • Each paragraph is too long, and it could be hard following the scheme logic all time. For this reason, I would like to suggest to authors to divide every paragraph in shorter subparagraph indicating the specific point described.

It was improved; paragraphs have been subdivided in shorter subparagraphs (with appropriate headlines). I hope that now are more readable and understandable.

  • Authors claim HBV and HCV as some risk factors for HCC. However, other pathologies such as cirrhosis could induce HCC into liver. Authors should add a briefly discussion about other pathologies.

It was improved; at the end of manuscript it has been added additional subparagraph described cirrhosis as a result of liver fibrosis in chronic liver disease, and primary biliary cirrhosis that could lead to HCC. Additionally subparagraph “Obesity related to leptin secretion is also a significant predictor of HCC in humans …” from some section was situated in this additional subparagraph (in green colour).

  • The difference in liver metabolism influenced the biliary tree metabolism and this could play a role in onset biliary diseases. Could the authors add a discussion about liver-tree metabolic link in this review?

It was done. To manuscript was added additional paragraph describing pathological condition associated with biliary tree in correlation with sex steroidal hormones.

Reviewer 2 Report

I am writing with references to manuscript number "ijerph-749640". This review is well written and I found one of the important update with relation to the gender-dependent HCC induction.

Thank you for considering me as a reviewer of this manuscript.

Author Response

Dear Reviewer,

Thank you for the revision of mine manuscript, and I am very glad that you liked it. Thank you very much for your appreciation.

Reviewer 3 Report

in the manuscript, ijerph-749640, the authors aimed to explore the molecular aspects of sex-hormone physiology and relate it with liver diseases.

In part, the authors have accomplished the objective, but in my opinion, the information they gathered is extraordinarily confusing and turns the reading harsh to understand.

it would be instructive if  the authors could provide more keywords.

In the first part of section 1, there is some confusion in biochemical aspects since glycogenolysis corresponds to the degradation of glycogen and is clearly different from glycolysis (enzymes, subproducts,...)

the part of PPAR gamma as a master regulator on insulin/glucose metabolism should be clearly explained since its one of the main functions is on lipid metabolism. These and many other molecular mechanisms should be considered and explained as expected in this type of manuscripts.

There are many other elegant forms to present information, and the authors should use images (through schematic figures) especially in section 1.

Author Response

Dear Reviewer,

Thank you for your effort to read and correct my manuscript. I would also like to thank you for your valuable comments. I have improved the manuscript and every changes have been marked in blue.

  • In part, the authors have accomplished the objective, but in my opinion, the information they gathered is extraordinarily confusing and turns the reading harsh to understand.

Because of that, it was improved, paragraph have been subdivided in shorter subparagraphs (with appropriate headlines). I hope that now are more readable and understandable. I hope this improvement finds your recognition.

  • it would be instructive if the authors could provide more keywords.

More keywords has been added. I hope this improvement finds your recognition.

  • In the first part of section 1, there is some confusion in biochemical aspects since glycogenolysis corresponds to the degradation of glycogen and is clearly different from glycolysis (enzymes, subproducts,...)

Because this is a Review Paper about gender-related disorders of hepatic metabolism, I did not focus on the biochemical (enzyme-dependent) glycogenolysis and glycolysis aspects, that is why this manuscript does not have this clearly explained, I hope that Readers of this publication (when will be publish) when they want it in their scope will explore this topic.

  • the part of PPAR gamma as a master regulator on insulin/glucose metabolism should be clearly explained since its one of the main functions is on lipid metabolism. These and many other molecular mechanisms should be considered and explained as expected in this type of manuscripts.

The sentence that “PPAR gamma as a master regulator on insulin/glucose metabolism ”was my mistake, that why it was removed, and because the information before this sentence contains the main view about regulation of the gene encoded GLUT2 by PPAR-γ. Moreover in the manuscript body there are the information about adipogenic function of PPAR-γ.

The mouse GLUT2 promoter includes both a PPAR-γ (peroxisome proliferator activator receptor-gamma) response element as well as a steroid hormone response element [45, 46], and that is why ERs with PPAR-γ could together regulate the gene encoded glucose transporter type [47]. As PPAR-γ (a master regulator of insulin signaling/glucose metabolism) is directly controlled by GLUT2, this receptor could play a role in the release of glucose from the liver [45].

  • There are many other elegant forms to present information, and the authors should use images (through schematic figures) especially in section 1.

Section 1 has been expanded/diversified with infographic that shown in simple way the processes such glycogenolysis and glycolysis. I hope this finds your recognition and approval.

Round 2

Reviewer 3 Report

In this revised version I've found improvements throughout the manuscript. In general, all sections have been changed.

I have some concerns regarding images. Was the figure 2 original? It seems a screenshot.